# A Nondestructive Indirect Approach to Long-Term Wood Moisture Monitoring Based on Electrical Methods

**DOI:** 10.3390/ma12152373

**Published:** 2019-07-25

**Authors:** Richard Slávik, Miroslav Čekon, Jan Štefaňák

**Affiliations:** 1Department of Wood Science and Technology, Faculty of Forestry and Wood Technology, 613 00 Brno, Czech Republic; 2Department of Physics, Faculty of Civil Engineering, Slovak University of Technology, 810 05 Bratislava, Slovakia; 3AdMaS Centre, Faculty of Civil Engineering, Brno University of Technology, 602 00 Brno, Czech Republic

**Keywords:** resistance measurement, wood moisture sensing, non-destructive testing, moisture safety

## Abstract

Wood has a long tradition of use as a building material due its properties and availability. However, it is very sensitive to moisture. Wood components of building structures basically require a certain level of moisture protection, and thus moisture monitoring to ensure the serviceability of such components during their whole lifespan while integrated within buildings is relevant to this area. The aim of this study is to investigate two moisture monitoring techniques promoting moisture safety in wood-based buildings (i.e., new structures, as well as renovated and protected buildings). The study is focused on the comparison of two electrical methods that can be employed for the nondestructive moisture monitoring of wood components integrated in the structures of buildings. The main principle of the two presented methods of the moisture measurement by electric resistance is based on a simple resistor–capacitor (RC) circuit system improved with ICM7555 chip and integrator circuit using TLC71 amplifier. The RC-circuit is easier to implement thanks to the digital signals of the used chip, whilst the newly presented integration method allows faster measurement at lower moisture contents. A comparative experimental campaign utilizing spruce wood samples is conducted in this relation. Based on the results obtained, both methods can be successfully applied to wood components in buildings for moisture contents above 8%.

## 1. Introduction

The properties and lifespan of wood strongly depend on many aspects, and therefore the permanent moisture monitoring of wood has a specific application in a wide range of technical fields. Primarily, wood-based wall technologies are widely considered to be suitable building materials for low-environmental impact composites in the building engineering field. Wood framed and composite wood wall technologies that utilize advanced insulation techniques have been well-integrated for several decades in the building sector, and specifically meet thermal and environmental requirements [1]. Thermal properties, such as thermal conductivity and specific heat, are given significant consideration when designing buildings, especially low-energy or passive ones. Building components that are exposed to the weather outdoors are mainly affected by moisture and temperature-related factors. It has been two decades since exterior building surfaces began to be designed to be monitored by methods for the continuous monitoring of temperature and moisture in the micro-environment of structures, and within the wood itself. A system was introduced that maps the climate index for the decay of wood at various geographical levels via the use of existing climatic data, standards, and moisture content measurements [2]. Moisture and temperature can also play an important role indoors and within the structure of a building, i.e., throughout the whole building envelope. The existence of high moisture content can initiate decay or the growth of fungi. Particularly for bio-based building materials such as wood, the use of biological agents should be considered in order to predict service life, particularly with regard to fungal decay and mold growth risks. The control and reduction of wood moisture content is therefore a key instrument for wood protection [3,4]. However, the permanent monitoring of wood components may also have a relevant role to play in this field, especially when such components are incorporated in the building structure and their continuous physical monitoring is impossible. The correct estimation of timber moisture content and the subsequent initiation of potentially necessary measures are therefore essential tasks during the planning, execution, and maintenance of buildings built with wood or wood-based products. This fact has contributed to a recent and considerable rise in interest concerning the in-situ monitoring of the moisture content of structural timber elements [5]. In this connection, the modeling of the outdoor performance of wood products is also attracting specific attention [6]. Furthermore, moisture content measurement has a lot of potential for use in testing the durability of timber products [7].

A technique for the nondestructive evaluation of moisture content distribution during drying using a newly developed soft X-ray digital microscope and absortimetry was investigated by Tanaka et al. [8] and Tanaka and Kawai [9]. X-ray-based methods and diagnostics have already been successfully developed and used in many applications to identify aspects of wood decay [10,11]. In particular, wood temperature and moisture content have a direct impact on fungus and its ability to metabolize and degrade wood cell wall material over time [12]. Moisture requirements for the growth and decay of different fungi and wood species have already been determined, though relationships between wood moisture content, wood temperature, and fungal decay play an important part when applying the method in specific climates [13]. In addition, the relationship between microclimate, material environment, and decay is being studied in order to achieve a better understanding of issues concerned with the service life prediction of wood and wood-based products. Dietsch et al. [5] describes common methods of determining wood moisture content and evaluates them with respect to their applicability in monitoring concepts. Continuous moisture measurements using calibrated load cells and a data logger coupled with a weather station are an efficient way to record moisture in all kinds of material [14]. Unfortunately, the most accurate direct methods, which use oven drying and distillation, are time-consuming and cause the destruction of specimens.

Many indirect methods have been developed based on electric conductance or dielectric properties which allow results to be obtained fast and with satisfactory accuracy. In this relation, wood moisture measurement has a long history. Indirect methods are often applied for in-situ measurements and monitoring. They are based on determining a different property, which is correlated with the water content. Thus, resistivity measurements require material-specific characteristics and a temperature compensation, since both parameters have a significant effect on electrical conductivity [15]. Dunlap [16] discusses twelve commercial electrical moisture meters. Most of them are based on resistance (conductance) measurement. The Wood Handbook [17] states that the resistance of wood ranges from a few petaohms for oven-dry wood to a few kiloohms for wood with fiber saturation. In the range between fiber saturation (appr. 30%) to complete saturation, the change of resistance is not so significant. James [18] mentioned that the conductance and dielectric properties of wood vary consistently with moisture content when it is less than 30%, with a roughly linear relationship between the logarithm of conductance and the logarithm of moisture content. Thanks to this relationship, the moisture content can be determined. The measurement principle is based on the application of direct or alternating current, but higher resistances than a few hundred megaohms are not so easy to measure. Electric-current through such huge resistance is often very small, and direct current measurement in a simple electrical circuit with an appropriate error is not possible. Measuring very high resistance values is a difficult task, since low voltage or currents are present and thus, noise and amplification must be carefully done, especially when low resistance values are required to be measured using the same circuit, too [19]. Moisture content is not the only phenomenon which has an effect on the conductance of wood material; another significant factor is temperature dependency. James [18] mentioned that an increase of 10 °C causes an approximately two-fold increase in conductance in regions with more than 10% moisture content. Another problem is the anisotropy of wood in the direction of the grain. Conductance measured parallel to the grain is approximately double that of perpendicular conductance.

Many methods, complex electrical circuits, and devices have been developed in past decades. The most common and most easily applicable methods are electrical resistance measurements [15]. Typical devices are equipped with probes and a display for showing calculated values. They are portable, and very useful for taking technical measurements or conducting on-site inspections of materials. A long-term moisture measuring and data logging method for wood exposed to weather conditions was developed by Brischke et al. [20]. The method involves measuring the electrical resistance with glued electrodes for a sustainable connection. The measuring points at the tips of the electrodes are glued conductively into the wood while the remaining outer parts of the electrodes are glued with insulating adhesive. For this purpose, special conductive and insulating glues and electrodes were developed and comparatively evaluated in laboratory tests. In a recent study, the comparison of accuracy and ease of operation between voltammetry and digital bridge method for electrical resistance measurement in wood specimens and the factors influencing voltammetry were examined in detail [21]. Another useful approach is based on interdigital capacitance sensing of moisture content in rubber wood where the electrode contact is non-penetrating. It has linear sensitivity and better accuracy at very low moisture content. This represents faster measurement and is more convenient to be used in the industry for the production line quality control monitoring of moisture content of rubber wood [22].

Nowadays, with innovative approaches in many fields, we often use technologies which allow us to monitor the service life of all building elements permanently. Portable independent devices often cannot be integrated into structures for a long time or connected to networks. For that reason, there are potential future applications for sensors which could be connected to a network and placed into a composite structure containing wood for a long period of time (Figure 1). The sophisticated circuits used in commercial portable devices are often trade secrets of their producers and are not so easy to adapt for use in automatized systems. If the resistance measurement approach is used, there are two methods which are applicable: They are subjected to further analysis here. Both of the analyzed methods use the capacitor charge principle in a different way.

This study describes experimental work which aimed to obtain wood moisture monitoring data indirectly using two electrical methods. The main aims of this research work are first, to introduce the theoretical principles of both measurement methods used, and second, to evaluate their applicability and finally to verify them based on the experimental results obtained. Both methods are applied to samples made of spruce wood. The experimental results were evaluated via a statistical approach based on Bland-Altman plots. Originally, the Bland-Altman procedure [23] was used in medicine to compare two clinical measurements that each produce some error in the data they measure [24]. Bland-Altman plots are now extensively used to evaluate the agreement between two different instruments or two measurement techniques. They allow the identification of any systematic difference between measurements, or possible outliers. The results of this statistical analysis method determine the applicability of both, monitoring methods and the overall consistency between them. The statistical analysis also identifies the moisture content range for which the methods should obtain acceptable results if properly applied.

## 2. Description of the Circuits, Theoretical Principles, and Applications

As mentioned above, there are basically two methods which can be used for long-term wood moisture monitoring using electrical resistance and direct current. The first method uses a resistor–capacitor (RC) circuit which has been improved by transforming it into a digital device using a 555 timer chip [25,26]. The second applicable method uses capacitor charging via an operational amplifier connected as an integrator circuit. This method was originally applied to other problems requiring the measurement of high resistance by Aguilar et al. [27]. Both methods can be usefully implemented in small sensor packages (Figure 1). It shows small sensor prototypes which were assembled based on this research work and directly attached to wood samples in a climatic chamber during a test involving long-term monitoring.

The application of circuits was tested on spruce wood samples. Several wood samples with electrodes attached in various orientations relative to the fibers of the wood were measured with both methods. The samples were fitted with electrodes and placed into a desiccator where humidity and temperature were controlled. The measurements were obtained via a digital oscilloscope at intervals during conditioning, and the weights of the samples were measured. The results from both circuits were compared to each other using the obtained data.

### 2.1. The RC Circuit Method

This type of circuit is also known as an RC network or RC filter. It combines a capacitor with a resistor in series and is driven by voltage or a source of current. The main principle of RC circuits is based on the relationship between capacitor and resistor values and the voltage level in the capacitor over time. In a simple example with direct current and a rise in voltage from zero to a particular voltage level at the input rail of the resistor, the time of capacitor charging is proportional to the time constant of the circuit. The time constant is defined by the value of the resistor and capacitor.

At zero time the capacitor *C* is discharged. If the input voltage is applied at resistor *R*, the voltage charge of capacitor *C* starts rising over time until it reaches the maximum value (Figure 2). The time constant (1) represents the time taken by the capacitor to charge to approximately 63.2% of its final voltage value. At a known and stable capacitance and a known time of charging the resistance can be estimated, and vice versa. The relation between the voltage of capacitor *V_C_* and the stable input voltage *V_IN_* on the left resistor rail can be described by Equation (2).

The described idea is applied in the first method. The charging and discharging of the circuit are managed by a timer-integrated circuit. The most well-known timer, the 555 timer chip, has many applications, from circuits with a blinking LED (Light-Emitting Diode) to circuits which use it to generate an AC (Alternating Current) signal. The use of this chip for wood moisture measurement was already mentioned by Vodicka [25]. It is a logical application in which the chip is used as a monostable timer. This simply means that if a trigger input is pulled to a logical zero, the capacitor is discharged, and then charging starts with the output of the chip set to a high logical level. If the voltage of the capacitor reaches 63% of the input voltage level, the output is set to the zero logical level.

The time of the output pulse is proportional to the RC constant and can be calculated with Equation (3). Testing was performed with an oscilloscope; the circuit (Figure 3) was triggered by a push button and time of charging was captured by oscilloscope (Figure 4). Triggering and time counting could be handled by a small microcontroller in a real world sensor. The problem standard timer chips have with leakage current, which is often higher than the current through a wood sample with a moisture content of under 10%, has been solved by the use of the ICM7555 chip. Its leakage current is within the range of a few picoamperes according the producer’s specifications.

(1)τ=RC [-]

(2)$$V_{C}=V_{in}\left(1-e^{\frac{-t}{\tau }}\right)\;[V]$$

(3)t=1.1RC [s]

### 2.2. The Integration Method

Operational amplifiers have various uses in electronics. The most common functions are as comparators, amplifiers, differentiators, and integrators. The last of these is the most relevant to high resistance measurement, and the second method implements operational amplifiers in that mode. A TLC71 amplifier in integrator mode has already been used to measure high resistance at the level of hundreds of megaohms [27]. Most operational amplifiers need a symmetric power supply, which is the main disadvantage of these devices in digital electronics.

The integrator works on the principle of capacitor charging. The voltage at the capacitor changes over time according to the voltage value applied to the negative rail of the amplifier (Figure 5).

The positive rail of the amplifier is grounded. If a negative voltage level is reached at input through the resistor, which represents measured resistance, the capacitor voltage grows to a positive supply voltage and vice versa. The slope of voltage change is proportional to the resistance and capacity. The output voltage is measured via an oscilloscope and the resetting of the integrator is triggered by a switch (Figure 6). In the case of a real-world sensor, this could be handled by a transistor managed by a microcontroller which is equipped with an analog to digital convertor for output voltage measurement.

## 3. Materials and Methods

An experimental investigation was carried out using the two abovementioned electrical resistance-based methods, and the measurement results were statistically evaluated. In the initial part of this research, the spruce wood samples were prepared, and test circuits were developed for their calibration using resistor fields. Then, the samples were conditioned, measured, and weighed on a continuous basis over three months. Finally, the obtained data were analyzed and statistically evaluated. Through this approach, a complex characterization of resistance could be comprehensively obtained from the measurements to verify the applicability of both methods used. All of the information on time-dependent moisture parameters measured from the samples was expected to show a clear picture of the evolution of moisture content can be provided if these test circuits are integrated within wood-based material or a real-world structure.

### 3.1. Sample Preparation Procedure

Three samples were prepared in the form of blocks with a cross section of 47.0 × 27.5 mm and a length of 90.0 mm (Figure 7). They were made of planed spruce wood, dried by electric oven at 105 °C and drilled with diameter 3 mm with deep 20 mm (Figure 7c).

### 3.2. Circuit Setup and Calibration Procedure

Test circuits were built on test printed circuit boards (PCBs) with through hole technology (THT) and surface-mount device (SMD) electronic parts. The trigger pulses for measurements were initialized by push buttons. The voltage response of the trigger and the output of the circuits were measured by a digital oscilloscope with a 25 MHz bandwidth. The RC circuit with a 555 timer chip used a 12 V single power supply, while the integrating circuit had a dual power supply with −12 V and +12 V rails. A polypropylene capacitor was chosen with the value 1 nF.

Both circuits were calibrated with two resistor fields. The first test field consisted of 15 resistors with 10 MΩ resistance, and the second test field had 50 resistors with 20 MΩ resistance. Each resistor was measured separately with two digital Ohmmeters and the results were averaged. The total resistance of the first field was set to 148.87 MΩ with a maximum error of 1.5 MΩ, and 993.4 MΩ with a maximum error of 9.9 MΩ. Several combinations of resistors were tested by both experimental circuits. The proportional constants of the circuits were identified according to the obtained data using statistical analysis (Figure 8).

### 3.3. Sample Conditioning and Measurement Procedure

The samples were conditioned in a desiccator above a solution of salt in accordance with technical standard EN 12751. Sodium chloride and potassium chloride were used in desiccator. Relative humidity at laboratory temperature (21 °C) reached approximately 76% with the first solution and about 86% with the second. The environment in the desiccator was monitored by a temperature and relative humidity probe (Figure 7b). The resistance and weight of the samples were measured at specified intervals by the circuits and a laboratory balance with a resolution of 0.01 g and an accuracy of 0.05 g. Each measurement was taken three times for each electrode location and the results were averaged. The whole conditioning period lasted for roughly three months.

The evolution of sample moisture content was identified by weighing, and the relative humidity in the desiccator was measured (Figure 9). Dry samples were placed into desiccator with solution of sodium chloride for first 1300 h. A logarithmic fast increase of moisture content at the beginning of conditioning can be identified, later continuing more slowly until finally resulting in its equilibrium. In the next step, after 200 h of equilibrium conditions, the samples were placed into another desiccator with higher partial pressure of water vapor. The vapor pressure slowly rises again to reach equilibrium, approaching 2500 h of measurement.

Resistance value of samples could be distorted by nonlinear distribution of moisture in depth form surface of a sample. The conditioning process takes a long time, however the expectation of quasi linear distribution of moisture in the sample is more realistic closer to states where equilibrium states are reached in desiccator.

### 3.4. Statistical Analysis

In the real world, every value measured at a given assembly or sensor often differs slightly, thanks to variations in the repeatability and accuracy of methods. The methods used in this research are no exception, and they can be expected to provide slightly varied results, sometimes with higher and sometimes with lower differences. The comparison of both methods requires statistical analysis, which was attempted via the application of the Bland-Altman procedure, which allows the identification of any systematic difference between measurements, as well as possible outliers. The mean difference is the estimated bias, and the standard deviation (SD) of the differences measures the random fluctuations around this mean. The 95% limits of agreement, i.e., ±1.96 SD of the difference, are computed to determine the most likely difference between two measurements conducted using two methods. If the differences within the ±1.96 SD are not physically important, the two methods may be used interchangeably. The 95% limits of agreement are often unreliable estimates of population parameters, especially for small sample sizes. For small sets of data, like those in the presented study, it is appropriate to use a two-sided 1 − (*α*/2) value of Student’s *t*-distribution with (*n* − 1) degree of freedom as a constant, which multiplies the standard deviation when calculating the limits of agreement.

## 4. Results

The resistance of the wood samples was obtained from the integration method circuit based on the slope of the rise in voltage. The resistance in megaohms was calculated based on data obtained from the calibration procedure. The obtained results for each electrode position differ slightly. Electrodes A and B obtained very similar values, but the values for electrode C were roughly double those measured by A and B (Table 1). This is in accordance with James [18]. Based on the proportional constant identified during the calibration procedure performed for the RC circuit, the resistance of samples was calculated. Positions A and B provided quite similar values, whilst those gained by position C were nearly two times higher (Table 2).

The relation between moisture content and calculated resistance has approximately linear behavior with resistance in logarithm scale (Figure 10, Figure 11 and Figure 12). Power regression lines have been added to the plots of individual measurements. The values obtained for each probe orientation have quite similar tendencies and sizes. Both methods reach very close values to each other in whole scale range of measurement. The influence of the probe orientation and the direction of the wood grain is about 2% (Figure 13). The curves obtained by the A and B probes are close to each other and lie below the curves obtained by the C probes, which are shifted horizontally by roughly around 2%. The measurement in perpendicular direction of fibers reached approximately two times higher value than those in parallel directions. It practically confirms general assumptions that are well covered in the literature. Relative conductivity values in the longitudinal, radial, and tangential directions are related by the approximate ratio of 1.0:0.55:0.50 [18].

## 5. Statistical Evaluation of Results

A study by the authors Bland and Altman [28] revealed that any two methods designed to measure the same parameter should show good correlation when a set of samples are chosen for which the parameter to be determined varies considerably. Therefore, a high correlation coefficient obtained for any two methods designed to measure the same property is just a sign that the sample chosen for measurement has a parameter which varies widely. It does not necessarily imply that there is a good agreement between the two methods. Hence, the analysis below was conducted to provide deeper insight into the differences between the two sets of measurements obtained by the two presented methods.

### 5.1. Construction of Bland-Altman Plots

The x axis represents the mean of the values measured by the RC method and the integration method. The y axis represents the differences between the values obtained by the two described methods. If there is agreement between two methods, the values in Figure 12 are expected to cluster around the mean of the differences (called the bias), and certainly within the limits of agreement. The dashed lines in the plots represent the lower and upper limit of agreement. In the presented case, only two points lie outside the limits of agreement.

### 5.2. Proportional Measurement Bias

Proportional bias can also be investigated via Bland-Altman plots, which indicate that the methods do not agree equally through the range of measurements. The limits of agreement are then dependent on the actual measurements. When the relationship between differences is identified, e.g., via regression analysis, the regression-based 95% limits of agreement should be provided, or proper transformation of the differences should be conducted. There is dependence in analyzed data between the mean values and differences. The Pearson´s correlation coefficient *r* = 0.85 is high. The found regression curve with regression coefficient R2 (Figure 14). In other words, the limits of agreement are underestimated (too wide) for low values and overestimated (too narrow) for high values. In such cases, logarithmic data transformation can be used. The aim of transformation is to determine the limits of agreement that are valid for the entire range of values. Logarithmically transformed data (resistances measured at position of electrodes A) are derived from the results (Figure 15). The x axis represents the logarithm of mean of the values measured by the RC method and the integration method. The y axis represents the logarithm of differences between the values obtained by both methods.

The regression coefficient decreased (R^2^ = 0.34) for the transformed data, and the points are more evenly distributed between the new limits of agreement. Nevertheless, three points still lie outside the limits of agreement. Logarithmic transformation was performed also for the two other data sets (resistances measured at position of electrodes B (Figure 16) and at position C (Figure 17), respectively). Two points corresponding to the moisture 7.33% and 7.66% are outside the limits of agreement in both these cases.

When the position of the electrode is ignored, there are still a few points out of the limits of agreements. Two of them are of importance, as they are even out of the outlier detector limits (dotted lines) calculated as mean ± 3 standard deviation (SD) (Figure 18).

## 6. Discussion

The main advantage of RC circuits is the simplicity of adapting timer circuits to emit digital signals. It is easier to integrate this kind of circuit in the sensors utilized by today’s digital electronic devices. The key limitation of such circuits is their long capacitor charge period at lower moisture contents. Sometimes, it may be necessary for a capacitor to charge for a relatively huge time; this would not be recorded by the controller, which evaluates the signal from the 555 timer chip at low moisture contents. The second problem is the resolution of charging time, which could be very small at higher moisture contents and, thus, could cause higher repeatability errors to affect measurements.

The integration method shows voltage changes, which allows the identification of lower moisture content much faster than an RC circuit. The main disadvantage of the integration method is the need for a symmetric power supply, as well as other circuits for the post-processing of analog signals. Post-processing circuits for analog signals are expensive and sensitive. The sensitivity of these circuits could lead to worse measurement repeatability and accuracy at lower moisture contents.

The range of differences between the two proposed methods is statistically important for small moisture contents of approximately below 8% (which correspond to high resistances), regardless of the fibers’ direction. At moisture contents under 8%, wood has very high resistance and these methods of measurement are affected by significant errors. Based on the results of the statistical analysis, it is recommended that the threshold moisture content for the use of the proposed methods be set at 8%. Both of the proposed methods show good agreement for a range of moisture contents above this threshold.

The threshold value of moisture content 8%, revealed by statistical analysis, is in line with findings of other authors. Papez et al. [29] performed moisture measurement by local resistivity sensors with a measurement range of 7–30%. The calibration curves were developed by Fernandez-Golfin et al. [30] for the estimation of ten hardwoods by means of electrical resistance measurement. All resistance measurements by Fernandez-Golfin et al. were taken by advanced laboratory tool AGILENT 4339B high resistance meter. They reported 8.0% as the lowest value of measurable moisture content.

The direct comparison with laboratory methods was not performed in this study. However, the uncertainty of the presented method can be deduced from comparison with other authors. Some of them, e.g., Fernandez-Golfin et al. [30] and Moron et al. [31], developed similar circuits and calibrated them with laboratory methods. Fernandez-Golfin et al. [30] predicted the moisture content of wood with an error ±1.0%. They also revealed that the measuring direction has negligible influence (error of estimation <±0.5%) on moisture content estimates, below the moisture content of about 15%. Moron et al. [31] compared results of moisture content measurement conducted by their capacitance meters with those based on the variation of electromagnetic transmittance of timber. They reported similar accuracy (below 1%) between both methods in measurement range 1 MΩ to 100 GΩ. Their results show how low power and low-cost circuits can be similar to high precision, cost, and size instruments.

## 7. Conclusions

Based on the data obtained, both tested methods—RC and integration method—can successfully identify moisture changes within the range of 8–15%, with a resolution and high accuracy of about 0.5% of moisture content. Furthermore, the advantages and disadvantages of the methods have been analyzed. The RC circuit is easier to implement thanks to the digital signals of the 555 chip, whilst the integration method allows faster measurement at lower moisture contents. The RC method seems to be more suitable for intelligent sensors integrated within the structure of a building to perform its long-term monitoring. Low moisture content significantly limits the effectiveness of the two methods due to the high resistance of the wood, and both methods can fail in this situation. At moisture contents higher than 8%, both methods seem to be adequately suitable. Based on an analysis of the results of both methods, moisture values ranging from 8–15% also reach similar levels from a statistical point of view (the differences between both methods are within the limits of agreement for this moisture content range). When two methods are compared to determine whether they can be used interchangeably, the ‘true’ value of the measured quantity is unknown. Hence, the comparability differs from calibration, which is the case when a true measure is compared with measurement by a new method. Linear regression is sometimes used inaccurately for comparison of two methods. A high correlation coefficient is just a sign that a widespread sample has been chosen to measure; however it does not imply that there is a good agreement between the two methods. The Bland-Altman technique was adopted for assessment of the magnitude of disagreement here, and based on the statistical analysis, the conformity of the measurement with the integrator and the RC circuit was provided. The accuracy of measurements with RC resistance is then supported by comparison with RC resistance that is already introduced by other authors [25,26]. Statistically, the relation between moisture content and calculated resistance has approximately linear behavior with resistance in logarithm scale. Furthermore, it was clearly demonstrated that the influence of the probe orientation and the direction of the wood grain is about 2%.

It is planned that further research will focus on the improvement of circuits and signal processing from an electrical point of view. Further measurements with higher moisture contents and various kinds of wood need to be analyzed using the two methods to expand the options for their application in buildings.

## Figures and Tables

**Figure 1 materials-12-02373-f001:**
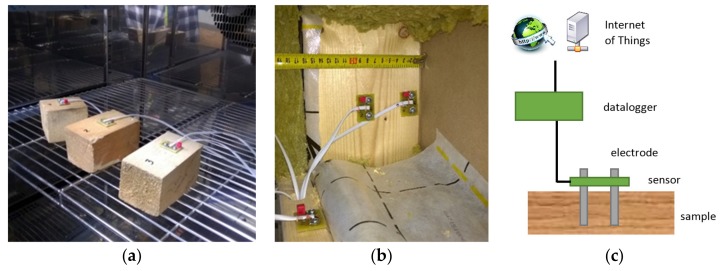
Implementation of small sensor module at real-world scale; (**a**) sensor testing in experimental chamber; (**b**) real wall structure integration; (**c**) functional scheme.

**Figure 2 materials-12-02373-f002:**
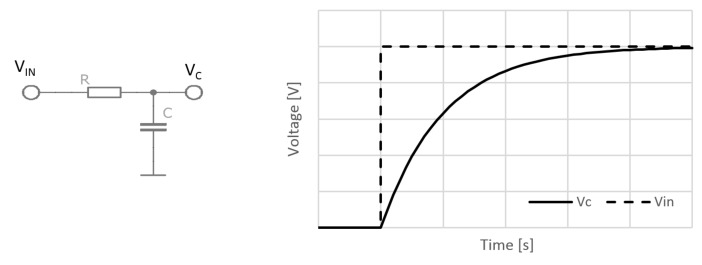
Resistor–capacitor (RC) circuit diagram and working principle.

**Figure 3 materials-12-02373-f003:**
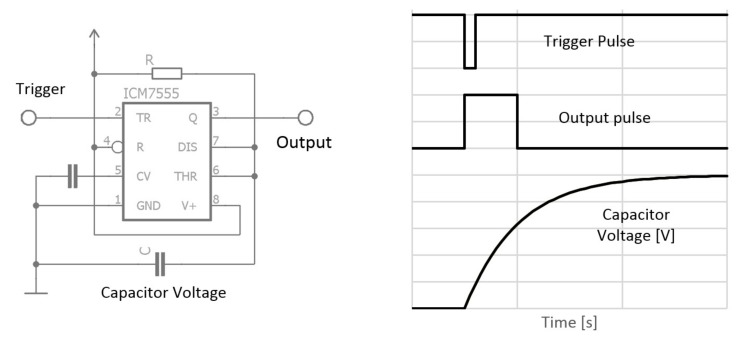
RC circuit improved by a 555 timer chip.

**Figure 4 materials-12-02373-f004:**
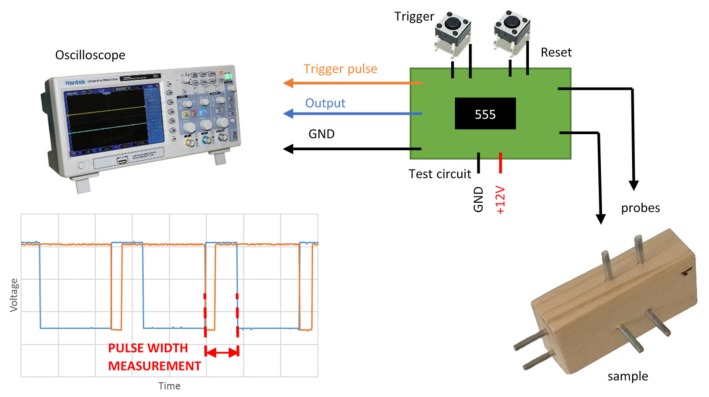
The complete principle of the RC circuit for the moisture content measurement.

**Figure 5 materials-12-02373-f005:**
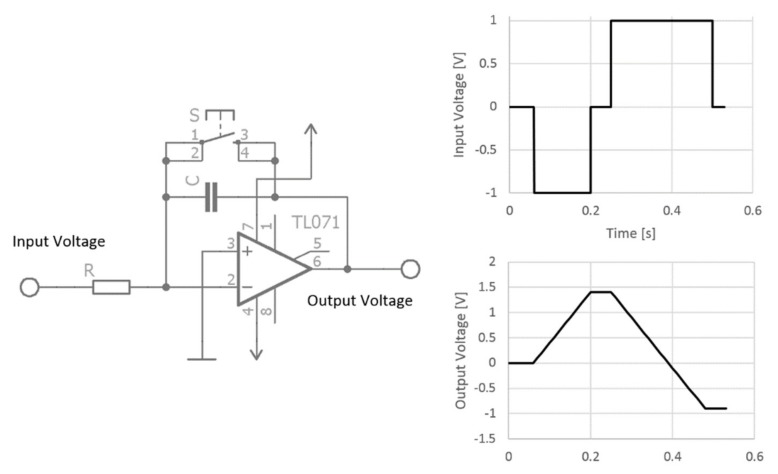
Integration method circuit diagram and working principle.

**Figure 6 materials-12-02373-f006:**
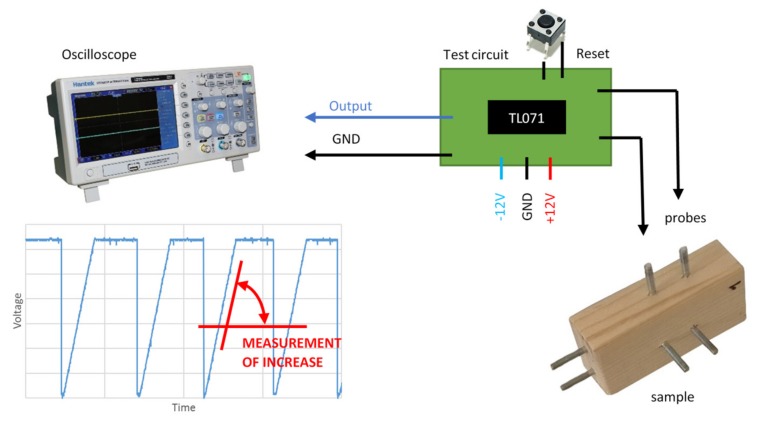
The complete principle of integration method for the moisture content measurement.

**Figure 7 materials-12-02373-f007:**
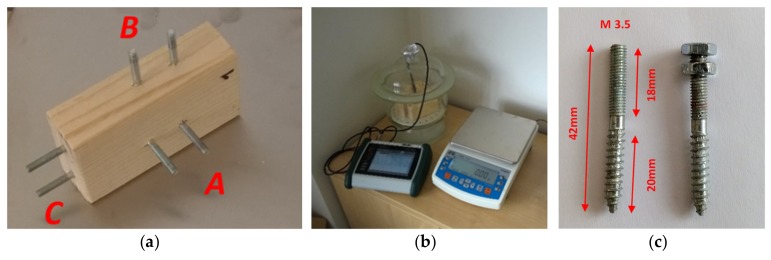
(**a**) Prepared test sample; (**b**) test conditioning; (**c**) detail of probes.

**Figure 8 materials-12-02373-f008:**
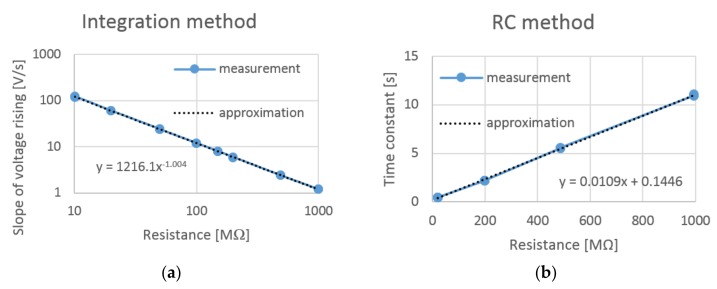
Calibration of (**a**) integrator and (**b**) RC circuit.

**Figure 9 materials-12-02373-f009:**
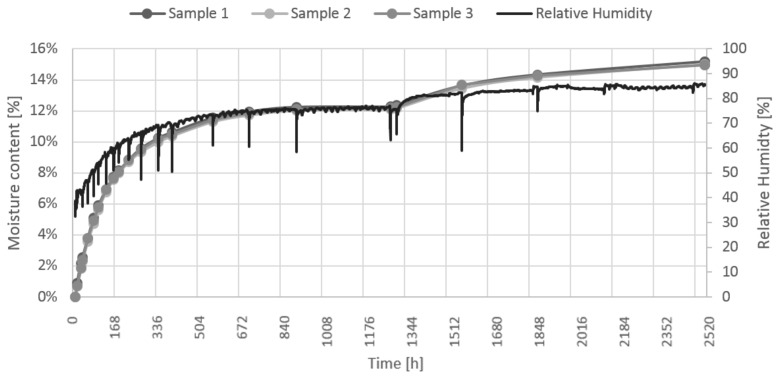
Evolution of moisture content over time.

**Figure 10 materials-12-02373-f010:**
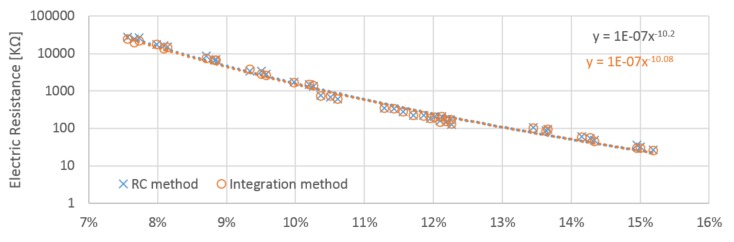
Obtained resistances from the integration and RC methods, the results calculated for the tops of the samples at the “A” electrodes.

**Figure 11 materials-12-02373-f011:**
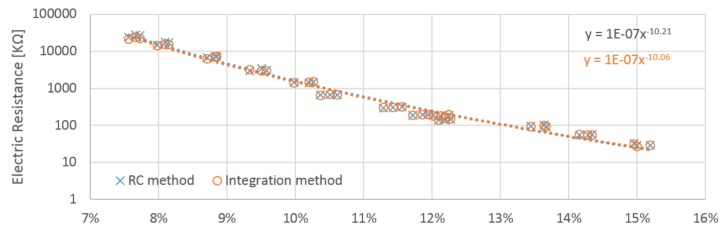
Obtained resistances from the integration and RC methods for the “B” probes placed on the right sides of the samples.

**Figure 12 materials-12-02373-f012:**
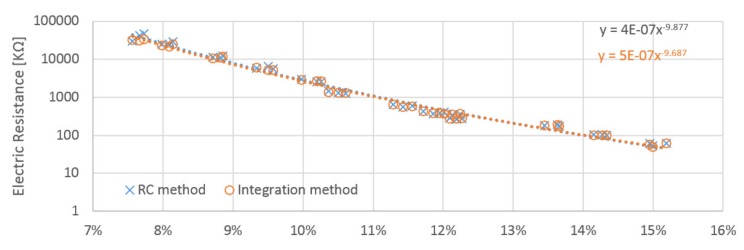
Obtained resistances from the integration and RC methods, the values calculated with the “C” probes placed in the cross sections.

**Figure 13 materials-12-02373-f013:**
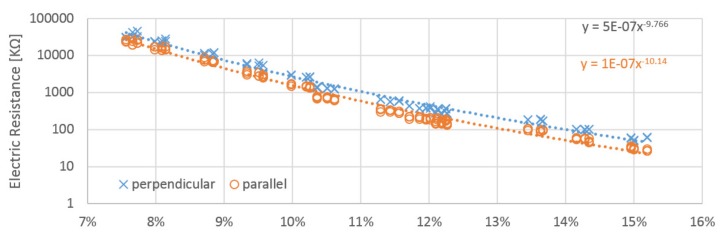
Obtained resistances from the integration and RC methods, the results calculated for the samples at the “A”, “B” and “C” electrodes together for parallel and perpendicular direction.

**Figure 14 materials-12-02373-f014:**
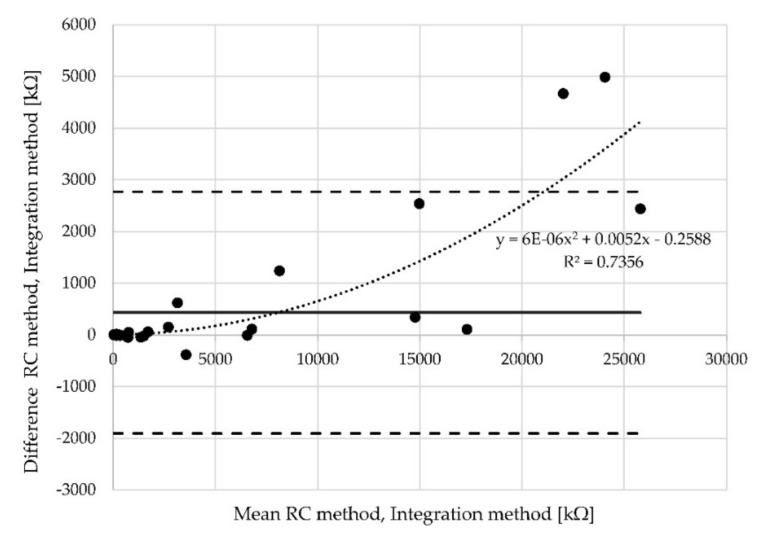
Bland-Altman plot for all samples measured at probe “A”.

**Figure 15 materials-12-02373-f015:**
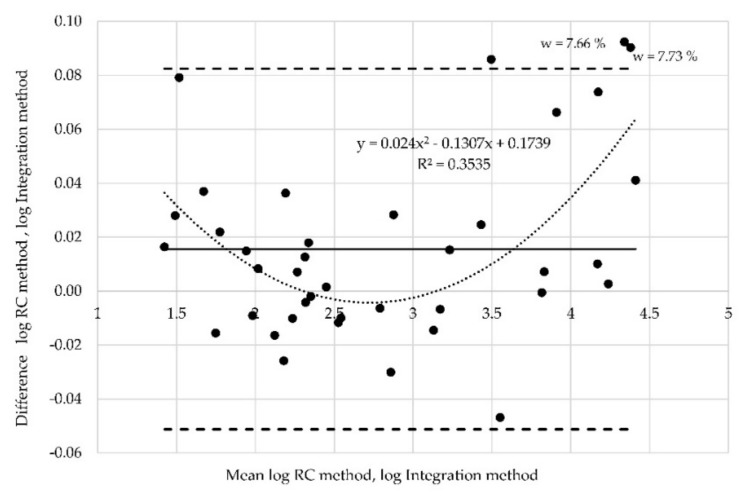
Altman plot for all samples measured at probe “A”—logarithmic transformation.

**Figure 16 materials-12-02373-f016:**
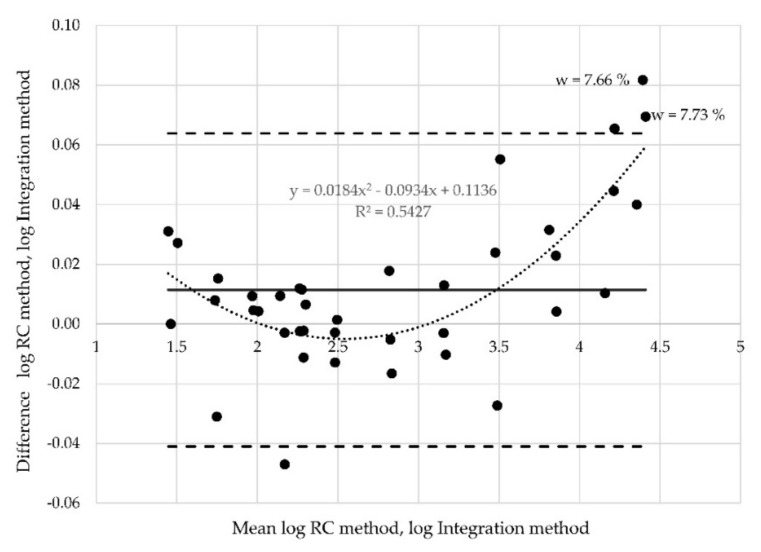
Altman plot for all samples measured at probe “B”—logarithmic transformation.

**Figure 17 materials-12-02373-f017:**
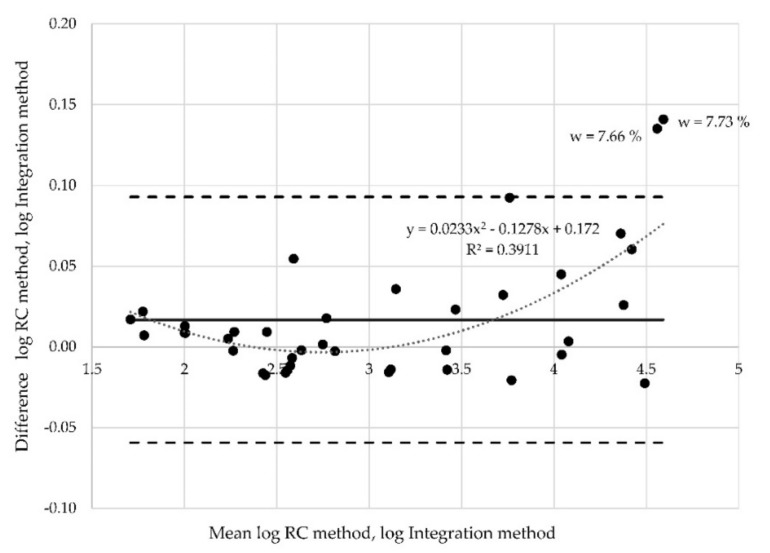
Altman plot for all samples measured at probe “C”—logarithmic transformation.

**Figure 18 materials-12-02373-f018:**
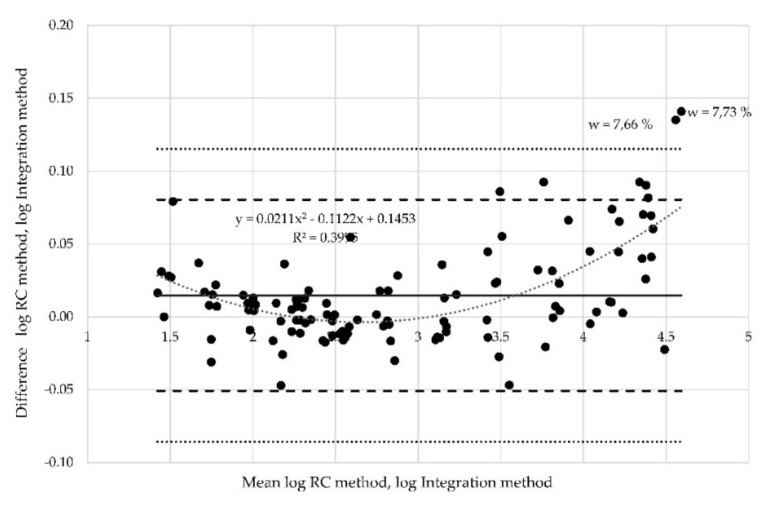
Bland-Altman plot for all samples and all probes—logarithmic transformation.

**Table 1 materials-12-02373-t001:** Calculated resistances from the RC method in megaohms (MOhm).

**Sample 1**													
	**7.73%**	**8.14%**	**8.85%**	**9.57%**	**10.26%**	**10.61%**	**11.56%**	**11.96%**	**12.25%**	**12.27%**	**13.63%**	**14.34%**	**15.19%**
A	26,550	14,936	6835	2775	1326	612	282	186	170	130	89	49	27
B	27,101	17,174	7220	3083	1461	665	313	201	191	147	102	58	29
C	46,009	28,257	12,046	5502	2601	1253	599	379	355	268	188	101	61
**Sample 2**													
	**7.56%**	**7.98%**	**8.71%**	**9.33%**	**9.98%**	**10.36%**	**11.29%**	**11.71%**	**12.02%**	**12.10%**	**13.45%**	**14.15%**	**15.00%**
A	27,009	17,358	8752	3372	1735	777	345	224	209	162	105	61	32
B	23,725	14,569	6716	2989	1462	672	299	191	185	140	95	54	29
C	30,183	24,550	11,514	5754	3009	1454	650	428	415	282	183	102	52
**Sample 3**													
	**7.66%**	**8.09%**	**8.83%**	**9.50%**	**10.20%**	**10.51%**	**11.43%**	**11.86%**	**12.13%**	**12.19%**	**13.66%**	**14.28%**	**14.95%**
A	24,349	16,239	6550	3453	1473	697	331	222	207	147	95	55	36
B	27,853	17,780	7321	3428	1427	669	302	193	183	140	94	55	33
C	42,294	24,936	10,945	6396	2599	1286	561	369	346	261	173	102	61

**Table 2 materials-12-02373-t002:** Calculated resistances from the integration method in MOhm.

**Sample 1**													
	**7.73%**	**8.14%**	**8.85%**	**9.57%**	**10.26%**	**10.61%**	**11.56%**	**11.96%**	**12.25%**	**12.27%**	**13.63%**	**14.34%**	**15.19%**
A	21,565	14,593	6723	2622	1371	621	281	183	174	135	86	45	26
B	22,452	15,499	7151	2918	1496	673	312	198	196	148	101	56	29
C	33,257	24,589	11,950	5109	2687	1299	575	385	367	279	184	99	60
**Sample 2**													
	**7.56%**	**7.98%**	**8.71%**	**9.33%**	**9.98%**	**10.36%**	**11.29%**	**11.71%**	**12.02%**	**12.10%**	**13.45%**	**14.15%**	**15.00%**
A	24,570	17,251	7513	3756	1675	728	353	225	203	149	103	58	30
B	21,639	14,227	6246	3183	1419	645	308	186	180	137	94	58	27
C	31,784	23,127	10,383	6034	2853	1339	654	430	366	276	184	100	50
**Sample 3**													
	**7.66%**	**8.09%**	**8.83%**	**9.50%**	**10.20%**	**10.51%**	**11.43%**	**11.86%**	**12.13%**	**12.19%**	**13.66%**	**14.28%**	**14.95%**
A	19,681	13,700	6558	2833	1496	747	340	213	209	156	97	57	30
B	23,737	15,293	6945	3019	1437	695	304	194	184	156	92	54	31
C	30,981	21,213	11,066	5170	2612	1328	559	379	359	271	171	99	58

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
