# Peer review of "A Nondestructive Indirect Approach to Long-Term Wood Moisture Monitoring Based on Electrical Methods"

_materials, 2019, doi:10.3390/ma12152373_

Round 1
Reviewer 1 Report
Paper can be accepted only after the following corrections:
- Figure 1 is not informative from scientific point of view. Please provide the schematic block diagram of measuring setup instead. Schematic block diagram of the sensor clarifying the idea would be also highly appreciated.
- Section 5 is valuable, however, I recommend to asses uncertainty of the method on the base of reference materials and moisture measurements by laboratory method.
- Conclusions should be stated in more quantitative way together with uncertainty assessment in accordance to the technical standards.
Author Response
Reviewer 1
Paper can be accepted only after the following corrections:
- Figure 1 is not informative from scientific point of view. Please provide the schematic block diagram of measuring setup instead. Schematic block diagram of the sensor clarifying the idea would be also highly appreciated.
Now, the Figure 1 is provided in a requested way.
- Section 5 is valuable, however, I recommend to asses uncertainty of the method on the base of reference materials and moisture measurements by laboratory method.
Thank you for your comment, the direct comparison with laboratory method was not performed in this study. However, the uncertainty of presented method can be deduced from comparison with other authors. See the last two paragraphs added into discussion section.
- Conclusions should be stated in more quantitative way together with uncertainty assessment in accordance to the technical standards.
In the revision, we’ve made the findings clear. The conclusion was modified.
Reviewer 2 Report
Dear Authors,
The manuscript:
A nondestructive indirect approach to long-term wood moisture monitoring based on electrical methods: experimental verification and statistical evaluation
is a very interesting paper and suitable for the journal.
The manuscript is well written, the structure is very good, text is understood and language is good.
Design of the actual experiment is very good, results are clear so conclusion.
My main concern of the paper is discussion chapter. Please add few paragraphs that present findings of other authors/papers and compare them with those obtained in your experiment.
There are also minor comments and corrections proposed in the main text that can be considered to improve the final version of the paper.
Congratulations to your paper, good luck in your further work.
With best wishes,
Reviewer

Author Response
Reviewer 2
The manuscript:
A nondestructive indirect approach to long-term wood moisture monitoring based on electrical methods: experimental verification and statistical evaluation is a very interesting paper and suitable for the journal.
The manuscript is well written, the structure is very good, text is understood and language is good.
Design of the actual experiment is very good, results are clear so conclusion.
My main concern of the paper is discussion chapter. Please add few paragraphs that present findings of other authors/papers and compare them with those obtained in your experiment.
New paragraphs are presented following your comment. In the revised text, we provided another discussion and paragraphs that put more details aiming to compare them with other papers/authors.
There are also minor comments and corrections proposed in the main text that can be considered to improve the final version of the paper.
Revised in a suggested way, thank you.
Congratulations to your paper, good luck in your further work.
Thank you
Reviewer 3 Report
The article is interesting. The subject of wood moisture and its measurement is an important matter. Humidity affect the properties and durability of wooden constructions.
The main topic of the article are new ideas of the moisture measurement by electric resistance using simple RC system system with ICM7555 chip and TLC71 amplifier.
Obtained results were statistical analysed and compared.
The methodology of tests is described unsatisfactorily. There is lack of information about how deep the probes are dipper in spruce wood.
There is no picture of real probes.
There are some language tissues in the text. There are also errors in the space between size and its value location. Somemistakes in table and figure captions.
The references are a bit old, there is co one published after 2015, it have to be updated.
In the text of the article, the remarks mentioned above and many others have been marked.

Author Response
Reviewer 3
The article is interesting. The subject of wood moisture and its measurement is an important matter. Humidity affect the properties and durability of wooden constructions.
The main topic of the article are new ideas of the moisture measurement by electric resistance using simple RC system system with ICM7555 chip and TLC71 amplifier.
Obtained results were statistical analysed and compared.
The methodology of tests is described unsatisfactorily. There is lack of information about how deep the probes are dipper in spruce wood.
Rewritten in a more informative way and new Figures are included to clarify it better.
There is no picture of real probes.
Now, Figure 7 presents the picture of real probes integrated.
There are some language tissues in the text. There are also errors in the space between size and its value location. Some mistakes in table and figure captions.
The text was revised, thank you.
The references are a bit old, there is co one published after 2015, it have to be updated.
According to your comments, in the revision, we updated a literature and several more references are included and cited in the Introduction section.
In the text of the article, the remarks mentioned above and many others have been marked.
We tried to respond on all mentioned remarks, thank you.
Reviewer 4 Report
The paper deals with a very topical issue as the need for use of wood as building and construction material increases resulting from the need to restrict CO2 emissions and increase the carbon sequestration efficiency. Therefore the control of the material durability and performance in use is of great importance. The research is well planned and carried out, and the results are logical and applicable for further development of moisture measuring techniques.
The title of the paper is comprehensive, but it could be shortened if possible.
The English language is mainly in order, but checking for remaining spelling errors is needed.
Some remarks can be found in the attachment.

Author Response
Reviewer 4
The paper deals with a very topical issue as the need for use of wood as building and construction material increases resulting from the need to restrict CO2 emissions and increase the carbon sequestration efficiency. Therefore the control of the material durability and performance in use is of great importance. The research is well planned and carried out, and the results are logical and applicable for further development of moisture measuring techniques.
The title of the paper is comprehensive, but it could be shortened if possible.
We made it shorten.
The English language is mainly in order, but checking for remaining spelling errors is needed. We provided a revision in that sense.
Some remarks can be found in the attachment.
We tried to respond on all mentioned remarks, thank you.
Reviewer 5 Report
Please find all my suggestions/commnets directly written into the pdf version.

Author Response
Reviewer 5
Please find all my suggestions/commnets directly written into the pdf version
We tried to respond on all mentioned remarks, thank you, following are the key comments.
The focus of the article is on resistance measurement, therefore in Intorduction part authors should discuss more in detail the influencing factors (temperature, moisture content, anatomical directions, etc...)
The third paragraph of the Introduction section presents the information about the influencing factors that specifically correspond to the ref 18, however new information was additionally added in this relation to refer more details about these factors.
The effect of UV light radiation must be discussed also, as the aesthetical value (Facades mainly) is very important! Sunlight contributes to surface degradation considerably (degr. of wood and degradation of surface finishing materials)
Thank you for your comment, we agree that all additional effects are important as well, however as the idea of this paper concerns primarily on the wood moisture monitoring of building elements that are integrated in building structures and components, the external effects are not directly considered in these cases, thus we decided not to include such effects.
Might the authors read this article and use the information to underline the relationship between wood's durability and moisture content.
Thank you for your comment, we included suggested reference in revised version.
"the whole psocess of evolution".... What these words convey to the readers? Too general sentence....
This was rewritten.
On Fig. 1. a climate chamber is shown with the samples. Line 142 refers to some desiccator. Which device was used? Chamber or desiccator. How the rel. humidity was regulated? Especially if the desiccator was used.
These relations are now explained and described in Section 3.3.
Important information is missing, how the samples reached the absolutely dry condition (visible on Fig. 7.)
This information is actually presented in the revised text.
It should be mentioned how deep the electrodes were placed into the wood. It is a critical info, as the moisture content in the samples will have some gradient.
Figure 7 is now extended with information regarding overall dimensions of electrodes applied.
If understood correctly, there were 3 stages during the tests as the moisture content was homogen in the samples (abs. dry, than equilibrium 1st and 2nd, ca. 12%, and ca. 15%). Authors later refers to different MCs during the measurements. In my opinion those values are not exact MC values, but some average values in depth, as there must be an MC gradient in depth. Still important data, but it has to be mentioned definitely in the text.
New paragraph in Section 3.3. is written to explain more better these relations that corresponds to Figure 9.
Round 2
Reviewer 1 Report
Paper was corrected and can be accepted in the present form.
Reviewer 3 Report
In my opinion aftetr revision the paper can be accepted in the present form.
There are few typesetting errors in the text.